# AutoJoin: Efficient Adversarial Training against Gradient-Free Perturbations for Robust Maneuvering via Denoising Autoencoder and Joint Learning

## Abstract

With the growing use of machine learning algorithms and ubiquitous sensors, many 'perception-to-control' systems are being developed and deployed. To ensure their trustworthiness, improving their robustness through adversarial training is one potential approach. We propose a gradient-free adversarial training technique, named AutoJoin, to effectively and efficiently produce robust models for image-based maneuvering. Compared to other state-of-the-art methods with testing on over 5M images, AutoJoin achieves significant performance increases up to the 40% range against perturbations while improving on clean performance up to 300%. AutoJoin is also highly efficient, saving up to 86% time per training epoch and 90% training data over other state-of-the-art techniques. The core idea of AutoJoin is to use a decoder attachment to the original regression model creating a denoising autoencoder within the architecture. This architecture allows the tasks 'maneuvering' and 'denoising sensor input' to be jointly learnt and reinforce each other's performance. The project code is at https://anonymous.4open.science/r/AutoJoin-FA13.

## 1 Introduction

The wide adoption of machine learning algorithms and ubiquitous sensors have together resulted in numerous tightly-coupled 'perception-to-control' systems being deployed in the wild. In order for these systems to be trustworthy, robustness is an integral characteristic to be considered in addition to their effectiveness. Adversarial training aims to increase the robustness of machine learning models by exposing them to perturbations that arise from artificial attacks (Goodfellow et al., 2014; Madry et al., 2017) or natural disturbances (Shen et al., 2021). In this work, we focus on the impact of these perturbations on image-based maneuvering and the design of efficient adversarial training for obtaining robust models. The test task is 'maneuvering through a front-facing camera'–which represents one of the hardest perception-to-control tasks since the input images are taken from partially observable, nondeterministic, dynamic, and continuous environments.

Inspired by the finding that model robustness can be improved through learning with simulated perturbations (Bhagoji et al., 2018), effective techniques such as AugMix (Hendrycks et al., 2019a), AugMax (Wang et al., 2021), MaxUp (Gong et al., 2021), and AdvBN (Shu et al., 2020) have been introduced for language modeling, and image-based classification and segmentation. The focus of these studies is not *efficient adversarial training for robust maneuvering*. AugMix is less effective to gradient-based adversarial attacks due to the lack of sufficiently intense augmentations; AugMax, based on AugMix, is less efficient because of using a gradient-based adversarial training procedure, which is also a limitation of AdvBN. MaxUp requires multiple forward passes for a single data point to determine the most harmful perturbation, which increases computational costs and time proportional to the number of extra passes.

Recent work by Shen et al. (2021) represents the state-of-the-art, gradient-free adversarial training method for achieving robust maneuvering against image perturbations. Their technique adopts Fréchet Inception Distance (FID) (Heusel et al., 2017) to determine distinct intensity levels of the perturbations that minimize model performance. Afterwards, datasets of single perturbations are generated. Before each round of training, the dataset that can minimize model performance is selected and incorporated with the clean dataset for training. A fine-tuning step is also introduced to boost model performance on clean images. While effective, examining the perturbation parameter space via FID adds complexity to the approach and using distinct intensity levels limits the model generalizability and hence robust efficacy. The approach also requires generating large datasets (2.1M images) prior to training, burdening computation and storage. Additional inefficiency and algorithmic complexity occur at training as the pre-round selection of datasets requires testing against perturbed datasets, resulting in vast data passing through the model.

We aim to develop a *gradient-free and efficient* adversarial training technique for robust maneuvering. Fig. 1 illustrates our approach, AutoJoin. We divide a steering angle prediction model into an encoder and a regression head. The encoder is attached by a decoder to form a denoising autoencoder (DAE). Using the DAE alongside the prediction model is motivated by the assumption that prediction on clean data is easier than on perturbed data. The DAE and the prediction model are jointly learnt: when perturbed images are forward passed, the reconstruction loss is added with the regression loss, enabling the encoder to simultaneously improve on 'maneuvering' and 'denoising sensor input.' AutoJoin enjoys efficiency as the extra computational cost stems only from passing the intermediate features through the decoder. AutoJoin is also easier to implement than Shen et al. (2021) as perturbations are randomly sampled within a moving range that is determined by linear curriculum learning (Bengio et al., 2009). The FID is used only minimally to determine the maximum intensity value of a perturbation. The model generalizability and robustness are significantly improved as more parameter space of the perturbation is explored, and 'denoising sensor input' provides the denoised training data for 'maneuvering.'

We test AutoJoin on four real-world driving datasets: Honda (Ramanishka et al., 2018), Waymo (Sun et al., 2020), Audi (Geyer et al., 2020), and SullyChen (Chen, 2017), which total over 5M clean and perturbed images and show AutoJoin achieves **the best performance on the maneuvering task while being the most efficient.** For example, AutoJoin outperforms Shen et al. (2021) up to **20% in accuracy and 43% in error reduction** using the Nvidia (Bojarski et al., 2016) backbone, and up to **44% error reduction** compared to other adversarial training techniques when using the ResNet-50 (He et al., 2016) backbone. AutoJoin is also highly efficient as it saves **8% per epoch time** compared AugMix (Hendrycks et al., 2019a) and saves **86% per epoch time and 90% training data** compared to Shen et al. (2021).

We provide extensive ablation studies to test the design of AutoJoin's pipeline. For example, we find that using all perturbations (discussed in Sec. 3.1) instead of a subset can avoid up to a 45% accuracy reduction and 51% error increase. We also find that not ensuring all perturbations are seen during learning and using distinct intensities from Shen et al. (2021), as opposed to random intensities, can cause up to a 16% error increase. Furthermore, we observe that incorporating the denoised images generated by the DAE into the training process leads to a decrease in accuracy by up to 10% and an increase in error by 42%.

## 2 Related Work

We next introduce techniques for improving model robustness against simulated image perturbations, and studies that use DAE to improve model robustness for driving.

Most adversarial training techniques against image perturbations to date have focused on image classification. To list some examples, AugMix (Hendrycks et al., 2019a) is a technique that enhances model robustness and generalizability by layering randomly sampled augmentations together. AugMax (Wang et al., 2021), a derivation of AugMix, trains on AugMix-generated images and their gradient-based adversarial variants. MaxUp (Gong et al., 2021) stochastically generates multiple augmented images of a single image and trains the

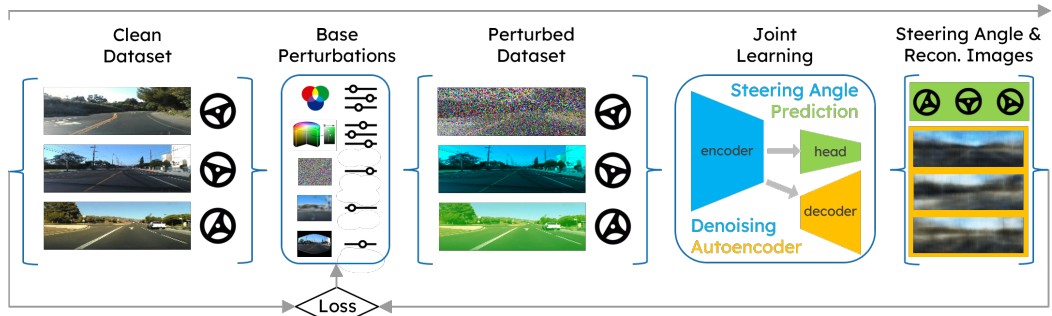

Figure 1: The pipeline of AutoJoin. The clean data comes from real-world driving datasets containing front-facing camera images and their corresponding steering angles. The perturbed data is prepared using various base perturbations and their sampled intensity levels. The steering angle prediction model and denoising autoencoder are jointly learnt to reinforce each other's performance. The resulting predictions and reconstructed images are used to compute the loss for adjusting perturbation intensity levels during learning.

model on the perturbed image that minimizes the model's performance. As a result, MaxUp requires multiple passes of a data point through the model for determining the most harmful perturbation. AdvBN (Shu et al., 2020) is a gradient-based adversarial training technique that switches between batch normalization layers based on whether the training data is clean or perturbed. It achieves state-of-the-art performance when used with techniques such as AugMix on ImageNet-C (Hendrycks and Dietterich, 2019).

Recently, Shen et al. (2021) has developed a gradient-free adversarial training technique against image perturbations. Their work uses Fréchet Inception Distance (FID) (Heusel et al., 2017) to select distinct intensity levels of perturbations. During training, the intensity that minimizes the current model's performance is adopted. While being the state-of-the-art method, the algorithmic pipeline combined with pre-training dataset generation are inefficient. First, an extensive analysis is needed to determine five intensity levels of perturbations. Second, the data selection process during training requires testing various combinations of perturbations and their distinct intensity levels. Third, significant costs are required for generating the pre-training datasets. In contrast, AutoJoin uses minimal FID analysis to obtain one point instead of five, which is then used for a range of intensities instead of having distinct intensities. AutoJoin also discards the mid-training data selection process in favor of ensuring all perturbations are seen each epoch. Lastly, AutoJoin generates perturbed datasets online during training.

DAEs have been used to improve model robustness for driving (Roy et al., 2018; Xiong and Zuo, 2022; Aspandi et al., 2019). For example, Wang et al. (2020) use an autoencoder to improve the accuracy of steering angle prediction by removing various roadside distractions such as trees or bushes. Their focus is not robustness against perturbed images as only clean images are used in training. DriveGuard (Papachristodoulou et al., 2021) explores different autoencoder architectures on adversarially degraded images that affect semantic segmentation rather than the steering task. They show that autoencoders can be used to enhance the quality of the degraded images, thus improving overall task performance. Xie et al. (2019) and Liao et al. (2018) use denoising as a method component to improve on their tasks' performance, where the focus is gradient-based attacks Chen et al. (2020), rather than gradient-free perturbations. The tasks are also restricted to classification instead of regression. Studies by Hendrycks et al. (2019b) and Chen et al. (2020) adopt self-supervised training to improve model robustness. However, their focus is again on (image) classification and not regression. To the best of our knowledge, our work is *the first gradient-free and efficient adversarial training technique for improving model robustness against perturbed image input in driving.*

## 3 METHODOLOGY

We show the pipeline of AutoJoin in Fig. 1. We use four driving datasets Honda (Ramanishka et al., 2018), Waymo (Sun et al., 2020), A2D2 (Geyer et al., 2020), and SullyChen (Chen,

2017)) in training and evaluating AutoJoin. During training, each image is perturbed by selecting a perturbation from a pre-determined set at a sampled intensity level (see Sec. 3.1). The perturbed images are then passed through DAE and the regression head for joint learning (see Sec. 3.2).

### 3.1 Image Perturbations and Intensity Levels

We use the same base perturbations as of Shen et al. (2021) to ensure fair comparisons. Specifically, we first perturb images' RGB color values and HSV saturation/brightness values in two directions, lighter or darker, according to a linear model: $v'_c = \alpha(a_c||b_c) + (1 - \alpha)v_c$, where $v'_c$ is the perturbed pixel value, $\alpha$ is the intensity level, $a_c$ is the channel value's lower bound, $b_c$ is the channel value's upper bound, and $v_c$ is the original pixel value. $a_c$ is used for the darker direction and has the default value 0 and $b_c$ is used for the lighter direction and has the default value 255. There are two exceptions: $a_c$ is set to 10 for the V channel to exclude a completely black image, and $b_c$ is set to 179 for the H channel according to its definition. The other base perturbations include Gaussian noise, Gaussian blur, and radial distortion, which are used to simulate natural corruptions to an image. The Gaussian noise and Gaussian blur are parameterized by the standard deviation of the image. Sample images of each base perturbation are shown in Appendix A.

In addition to the nine base perturbations, the channel perturbations (i.e., R, G, B, H, S, V) are further discretized into their lighter or darker components such that if $p$ is a channel perturbation, it is decomposed into $p_{light}$ and $p_{dark}$. As a result, the perturbation set contains 15 elements. During learning, we expose the model to all 15 perturbations (see Algorithm 1) with the aim to improve its generalizability and robustness. We refer to using the **F**ull **S**et of 15 perturbations as FS in Sec. 4. AutoJoin is trained on images with single perturbations, yet proves effective not only on such images but also on those with multiple and unseen perturbations.

The intensity level of a perturbation is sampled within $[0, c)$. The minimum 0 represents no perturbation, and $c$ is the current maximum intensity. The range is upper-bounded by $c_{max}$, whose value is inherited from Shen et al. (2021) to ensure comparable experiments. In practice, we scale $[0, c_{max}]$ to $[0, 1)$. After each epoch of training, $c$ is increased by 0.1 providing the model loss has reduced comparing to previous epochs. The entire training process begins on clean images ($[0, 0)$). In contrast to Shen et al., our approach allows the model to explore the entire parameter space of a perturbation (rather than on distinct intensity levels). We refer to using **R**andom **I**ntensities as RI in Sec. 4. Further exploration of the perturbation parameter space by altering the minimum/maximum values is discussed in Appendix B.

### 3.2 Joint Learning

The denoising autoencoder (DAE) and steering angle prediction model are jointly learnt. The DAE learns how to denoise the perturbed sensor input, while the prediction model learns how to maneuver given the denoised input. Both models train the shared encoder's latent representations, resulting in positive transfer between the tasks for two reasons. First, the DAE trains the latent representations to be the denoised versions of perturbed images, which enables the regression head to be trained on denoised representations rather than noisy representations, which may deteriorate the task performance. Second, the prediction model trains the encoder's representations for better task performance, and since the DAE uses these representations, the reconstructions are improved in favoring the overall task.

Our approach is formally described in Algorithm 1. For a clean image $x_i$, a perturbation and its intensity $l \in [0, c)$ are sampled. The augmented image $y_i$ is a function of the two and is passed through the encoder $e(\cdot)$ to obtain the latent representation $z_i$. Next, $z_i$ is passed through both the decoder $d(\cdot)$ and the regression model $p(\cdot)$, where the results are the reconstruction $x'_i$ and steering angle prediction $a_{p_i}$, respectively. Every 15 images, the perturbation set is randomized to prevent overfitting to a specific pattern of perturbations.

---

**Algorithm 1** AutoJoin

---

**input:** training batch $\{x_i\}_n$ (clean images), encoder $e$, decoder $d$, regression model $p$, perturbations $\mathcal{M}$, curriculum bound $c$
**for each** epoch **do**
    **for each** $i \in 1,...,n$ **do**
        Select perturbation $op = \mathcal{M}[i \bmod len(\mathcal{M})]$
        Sample intensity level $l$ from $[0, c)$ randomly
        $y_i = op(x_i, l)$ // perturb a clean image
        $z_i = e(y_i)$ // obtain the latent representation
        $x'_i = d(z_i)$ // reconstruct an image from the latent representation
        $a_p = p(z_i)$ // predict a steering angle using the latent representation
        **if** $i \% len(\mathcal{M}) = 0$ **then**
            Shuffle $\mathcal{M}$ // randomize the order of perturbations
        **end if**
    **end for**
    Calculate $\mathcal{L}$ using Eq. 1
    **if** $\mathcal{L}$ improves **then**
        Increase $c$ by 0.1 // increase the curriculum's difficulty
    **end if**
    Update $e$, $d$, and $p$ to minimize $\mathcal{L}$
**end for**
**return** $e$ and $p$ // for steering angle prediction

---

For the DAE, the standard $\ell_2$ loss is used by comparing $x'_i$ to $x_i$. For the regression loss, $\ell_1$ is used between $a_{p_i}$ and $a_{t_i}$, where the latter is the ground truth angle. The two losses are combined for the joint learning:

$$\mathcal{L} = \lambda_1 \ell_2 \left( \mathbf{x'_i}, \mathbf{x_i} \right) + \lambda_2 \ell_1 \left( \mathbf{a_{p_i}}, \mathbf{a_{t_i}} \right). \tag{1}$$

The weights $\lambda_1$ and $\lambda_2$ are set as follows. For the experiments on the Waymo (Sun et al., 2020) dataset, $\lambda_1$ is set to 10 and $\lambda_2$ is set to 1 for better performance (emphasizing reconstructions). For the other three datasets, $\lambda_1$ is set to 1 and $\lambda_2$ is set to 10 to ensure the main focus of the joint learning is 'maneuvering.' Once training is finished, the decoder is detached, leaving the prediction model for testing through datasets in six categories (see Sec. 4.1 for details).

## 4 EXPERIMENTS AND RESULTS

### 4.1 EXPERIMENT SETUP

We compare AutoJoin to five other approaches: Shen et al. (2021) (referred to as Shen hereafter), AugMix (Hendrycks et al., 2019a), MaxUp (Gong et al., 2021), AdvBN (Shu et al., 2020), and AugMax (Wang et al., 2021). We also compare to a Standard model, one trained using only clean images, and a Standard model trained with the **F**ull **S**et of 15 perturbations and **R**andom **I**ntensities discussed in Sec. 3.1. We test on two backbones, the Nvidia model (Bojarski et al., 2016) and ResNet-50 (He et al., 2016). The breakdown of the two backbones, training parameters, and computing platforms are detailed in Appendix A. We use four driving datasets in our experiments: Honda (Ramanishka et al., 2018), Waymo (Sun et al., 2020), A2D2 (Geyer et al., 2020), and SullyChen (Chen, 2017). They have been widely adopted for developing machine learning models for driving-related tasks (Xu et al., 2019; Shi et al., 2020; Yi et al., 2021; Shen et al., 2021). Based on these four datasets, we generate test datasets according to Shen to ensure fair comparisons. The test datasets contain more than 5M images in four categories: Clean, Single, Combined, and Unseen. Combined contains images each with several single perturbations overlaid and Unseen contains unobserved images with perturbations extracted from the ImageNet-C dataset Hendrycks and Dietterich (2019). Sample images for each category are given in Fig. 2. The details of these datasets and more sample images are given in Appendix A. We evaluate our approach using mean accuracy (MA) and mean absolute error (MAE), whose definitions are also given in Appendix A.

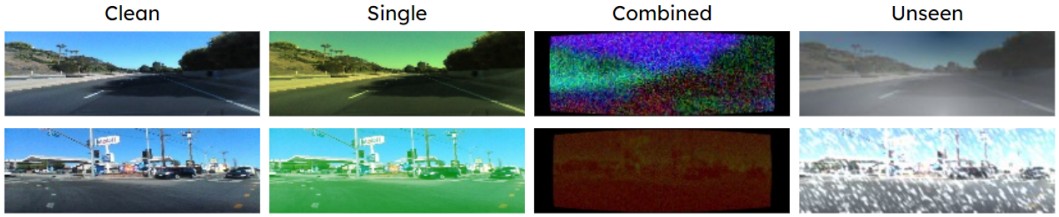

Figure 2: Sample images for each category. For Single, from top to bottom, the perturbations applied are B Lighter and G Darker, respectively. Combined consists of six test sets with images from Set 1 (top) and Set 6 (bottom) given. Fog and snow perturbations are shown for Unseen.

Table 1: Results on the SullyChen dataset using the Nvidia backbone. AutoJoin outperforms all other techniques in all test categories and improves the clean performance three times over Shen when compared to Standard.

|  | Clean | | Single | | Combined | | Unseen | |
| --- | --- | --- | --- | --- | --- | --- | --- | --- |
|  | MA (%) | MAE | MA (%) | MAE | MA (%) | MAE | MA (%) | MAE |
| Standard | 86.19 | 3.35 | 66.19 | 11.33 | 38.50 | 25.03 | 67.38 | 10.94 |
| Standard (FSRI) | 78.21 | 4.73 | 74.91 | 5.97 | 61.54 | 11.93 | 71.98 | 7.24 |
| AdvBN | 79.51 | 5.06 | 69.07 | 9.18 | 44.89 | 20.36 | 67.97 | 9.78 |
| AugMix | 86.24 | 3.21 | 79.46 | 5.21 | 49.94 | 17.24 | 74.73 | 7.10 |
| AugMax | 85.31 | 3.43 | 81.23 | 4.58 | 51.50 | 17.25 | 76.45 | 6.35 |
| MaxUp | 79.15 | 4.40 | 77.40 | 5.01 | 61.72 | 12.21 | 73.46 | 6.71 |
| Shen | 87.35 | 3.08 | 84.71 | 3.76 | 53.74 | 16.27 | 78.49 | 6.01 |
| AutoJoin | **89.46** | **2.86** | **86.90** | **3.53** | **64.67** | **11.21** | **81.86** | **5.12** |

## 4.2 RESULTS

The main results are discussed in Sec. 4.2.1. The elements of AutoJoin are examined in Sec. 4.2.2. The efficiency of AutoJoin is introduced in Sec. 4.2.3. The results reported are the averages over all test cases of a given test category.

### 4.2.1 EFFECTIVENESS AGAINST GRADIENT-FREE PERTURBATIONS

Table 1 shows the comparison results on the SullyChen dataset using the Nvidia backbone. AutoJoin *outperforms every other adversarial technique across all test categories* in both performance metrics. In particular, AutoJoin improves accuracy on Clean by 3.3% MA and 0.58 MAE compared to the standard model trained solely on clean data. This result is significant as the clean performance is the most difficult to improve while AutoJoin achieves about **three times the improvement** on Clean compared to the state-of-the-art performance of Shen. Tested on the perturbed datasets, AutoJoin achieves 64.67% MA on Combined – a **20% accuracy increase** compared to Shen, 11.21 MAE on Combined – a **31% error decrease** compared to Shen, and 5.12 MAE on Unseen – another **15% error decrease** compared to Shen.

Table 2 shows the comparison results on the A2D2 dataset using the Nvidia backbone. AutoJoin again *outperforms all other techniques*. To list a few notable improvements over Shen: 6.7% MA improvement on Clean to the standard model, which is a **4.2% performance increase**; 11.72% MA improvement – a **17% accuracy increase**, and 6.48 MAE drop – a **43% error decrease** on Combined; 5.01% MA improvement – a **7% accuracy increase** and 1.76 MAE drop – a **8% error decrease** on Unseen.

Switching to the ResNet-50 backbone, Table 3 shows the results on the Honda dataset. Here, we only compare to Shen and AugMix because Shen is the latest technique and AugMix was the state-of-the-art before Shen, which also has the ability to improve both clean and robust performance on driving datasets. As a result, AutoJoin *outperforms both Shen and AugMix on perturbed datasets in most categories*. Specifically, AutoJoin achieves **the highest MAs across all perturbed categories**. AutoJoin also drops the MAE to

Table 2: Results on the A2D2 dataset using the Nvidia backbone. AutoJoin outperforms every other approach in all test categories while improving on Clean by a wide margin of 4.2% MA compared to Shen, achieving 168% performance increase.

|  | Clean | | Single | | Combined | | Unseen | |
|---|---|---|---|---|---|---|---|---|
|  | MA (%) | MAE | MA (%) | MAE | MA (%) | MAE | MA (%) | MAE |
| Standard | 78.00 | 8.07 | 61.51 | 21.42 | 43.05 | 28.55 | 59.41 | 26.72 |
| Standard (FSRI) | 77.95 | 8.33 | 77.31 | 8.64 | 72.45 | 10.39 | 74.04 | 10.49 |
| AdvBN | 76.59 | 8.56 | 67.58 | 12.41 | 43.75 | 24.27 | 70.64 | 11.76 |
| AugMix | 78.04 | 8.16 | 73.94 | 10.02 | 58.22 | 20.66 | 71.54 | 11.44 |
| AugMax | 77.21 | 8.79 | 75.14 | 10.43 | 60.81 | 23.87 | 72.74 | 11.87 |
| MaxUp | 78.93 | 8.17 | 78.36 | 8.42 | 71.56 | 13.22 | 76.78 | 9.24 |
| Shen | 80.50 | 7.74 | 78.84 | 8.32 | 67.40 | 15.06 | 75.30 | 9.99 |
| AutoJoin | **84.70** | **6.79** | **83.70** | **7.07** | **79.12** | **8.58** | **80.31** | **8.23** |

Table 3: Results of comparing AutoJoin to AugMix and Shen on the Honda dataset using ResNet-50. AutoJoin achieves the best overall robust performance. However, Shen's fine-tuning stage solely on Clean images grants them an advantage on Clean.

|  | Clean | | Single | | Combined | | Unseen | |
|---|---|---|---|---|---|---|---|---|
|  | MA (%) | MAE | MA (%) | MAE | MA (%) | MAE | MA (%) | MAE |
| Standard | 92.87 | 1.63 | 73.12 | 11.86 | 55.01 | 22.73 | 69.92 | 13.65 |
| Standard (FSRI) | 88.58 | 2.27 | 86.11 | 3.30 | 47.85 | 39.12 | 81.93 | 4.92 |
| AugMix | 90.57 | 1.97 | 86.82 | 3.53 | 64.01 | 15.32 | 84.34 | 4.31 |
| Shen | **97.07** | **0.93** | 93.08 | 2.52 | 70.53 | **13.20** | 87.91 | 4.94 |
| AutoJoin | 96.46 | 1.12 | **94.58** | **1.98** | **70.70** | 14.56 | **91.92** | **2.89** |

1.98 on Single, achieving **44% improvement** over AugMix and **21% improvement** over Shen; and drops the MAE to 2.89 on Unseen, achieving **33% improvement** over AugMix and **41% improvement** over Shen. On this particular dataset, Shen outperforms AutoJoin on Clean by small margins due to its additional fine-tuning step on Clean. Nevertheless, AutoJoin still manages to improve upon the standard model and AugMix on Clean by large margins.

During testing, we find Waymo to be unique in that the model benefits more from learning the inner representations of the denoised images. Therefore, we slightly modify the procedure of Algorithm 1 after perturbing the batch as follows: 1) one-tenth of the perturbed batch is sampled; 2) for each single perturbed image sampled, two other perturbed images are sampled; and 3) the three images are averaged to form a 'fused' image. This is different from AugMix as AugMix applies multiple perturbations to a single image. We term this alternative procedure AutoJoin-Fuse.

Table 4: Results of comparing our approaches (AutoJoin and AutoJoin-Fuse) to AugMix and Shen on the Waymo dataset using ResNet-50. Our approaches not only improve on Clean the most, but also achieve the best overall robust performance.

|  | Clean | | Single | | Combined | | Unseen | |
|---|---|---|---|---|---|---|---|---|
|  | MA (%) | MAE | MA (%) | MAE | MA (%) | MAE | MA (%) | MAE |
| Standard | 61.83 | 19.53 | 55.99 | 31.78 | 45.66 | 55.81 | 57.74 | 24.22 |
| Standard (FSRI) | 61.83 | 20.15 | 61.45 | 20.29 | 56.95 | 24.94 | 60.56 | 21.06 |
| AugMix | 61.74 | 19.19 | 60.83 | 20.10 | 56.34 | 24.23 | 59.78 | 21.75 |
| Shen | 64.77 | 18.01 | 64.07 | 19.77 | 61.67 | **20.28** | 63.93 | 18.77 |
| AutoJoin | 64.91 | 18.02 | 63.84 | 19.30 | 58.74 | 26.42 | 64.17 | 19.10 |
| AutoJoin-Fuse | **65.07** | **17.60** | **64.34** | **18.49** | **63.48** | 20.82 | **65.01** | **18.17** |

Table 5: Results of comparing AutoJoin with or without DAE on the SullyChen dataset with the Nvidia backbone. Using DAE allows AutoJoin to achieve three times the performance gain on Clean over Shen. AutoJoin without DAE performs worse than Shen on Clean and Single MAE. Ours without FSRI performs worse than Shen, but better than AugMix. These changes result in our method performing worse than Shen, showing the necessity of AutoJoin's pipeline design.

| | Clean | | Single | | Combined | | Unseen | |
|---|---|---|---|---|---|---|---|---|
| | MA (%) | MAE | MA (%) | MAE | MA (%) | MAE | MA (%) | MAE |
| Standard | 86.19 | 3.35 | 66.19 | 11.33 | 38.50 | 25.03 | 67.38 | 10.94 |
| AugMix | 86.24 | 3.21 | 79.46 | 5.21 | 49.94 | 17.24 | 74.73 | 7.10 |
| Shen | 87.35 | 3.08 | 84.71 | 3.76 | 53.74 | 16.27 | 78.49 | 6.01 |
| Ours, w/o DAE | 88.30 | 3.09 | 85.75 | 3.81 | 62.96 | 11.90 | 81.09 | 5.33 |
| Ours, w/o FSRI | 86.43 | 3.54 | 83.19 | 4.62 | 61.97 | 13.01 | 78.51 | 6.23 |
| Ours (AutoJoin) | **89.46** | **2.86** | **86.90** | **3.53** | **64.67** | **11.21** | **81.86** | **5.12** |

Table 4 shows the results on the Waymo dataset using ResNet-50. AutoJoin-Fuse makes a prominent impact by *outperforming Shen on every test category except for combined MAE*. We also improve the clean performance over the standard model by **3.24% MA and 1.93 MAE**. AutoJoin also outperforms AugMix by margins up to **7.14% MA and 3.41 MAE**. These results are significant as for all four datasets, the well-performing robust techniques operate within 1% MA or 1 MAE.

While not the focus of this project, we additionally explore AutoJoin's effectiveness against gradient-based adversarial examples. The results and discussion are provided in Appendix G.

### 4.2.2 EFFECTIVENESS OF AUTOJOIN PIPELINE

**DAE and Feedback Loop**. A major component of AutoJoin is DAE. The results of our approach with or without DAE are shown in Table 5. AutoJoin without DAE outperforms Shen in several test categories but not on Clean and Single MAE, meaning the perturbations and sampled intensity levels are effective for performance gains. In addition, a byproduct of DAE is denoised images. A natural idea is to use these images as additional training data for the prediction model, thus forming a feedback loop within AutoJoin. We explore the feedback loop in greater detail in Appendix C. Overall, our experiments show a decrease in performance due to the feedback loop, thus we exclude it from the AutoJoin's pipeline.

**Perturbations and Intensities**. To better understand the base perturbations, we conduct experiments with six different perturbation subsets: 1) No RGB perturbations, 2) No HSV perturbations, 3) No blur, Gaussian noise, or distort perturbations (denoted as BND), 4) only RGB perturbations and Gaussian noise, 5) only HSV perturbations and Gaussian noise, and 6) no blur or distort perturbations. These subsets are formed to examine the effects of the color spaces and/or blur, distort, or Gaussian noise. We exclude Single from the results, shown in Table 6 as different subsets will cause Single becoming a mixture of unseen and seen perturbations. Not using blur or distort outperforms using all perturbations within Combined by 1.11 MA and 0.33 MAE, but not within Clean and Unseen. We observe that using RGB perturbations tends to result in better Clean performance. Not using the HSV perturbations results in the worse generalization performance out of the models with 50.22% MA and 78.91% MA in Combined and Unseen, respectively. Overall, we find that using all 15 perturbations is necessary for maximal performance. More results and findings for the other driving datasets is given in Appendix E.

We also assess AutoJoin's performance without using the **F**ull **S**et of 15 perturbations and **R**andom **I**ntensities (FSRI), which are described as AutoJoin without FSRI in Table 5. In this case, perturbations are randomly sampled during the learning process, and Shen's distinct intensity values are adopted. The results in Table 5 show a significant decrease in performance when FSRI are not used for AutoJoin. Thus, we conclude that both FS and RI are necessary for optimal performance and to surpass Shen. More results and details on the FS and RI components for different datasets are provided in Appendix D.

Table 6: Results on the SullyChen dataset with the Nvidia backbone using six subsets of the base perturbations. 'w/o BND' means no presence of blur, noise, and distort. Single is removed for a fair comparison. Using all base perturbations results in the best overall performance.

| | Clean | | Combined | | Unseen | |
|---|---|---|---|---|---|---|
| | MA (%) | MAE | MA (%) | MAE | MA (%) | MAE |
| w/o RGB | 87.71 | 3.00 | 58.88 | 14.71 | 81.23 | 5.13 |
| w/o HSV | 88.33 | 2.91 | 50.22 | 18.05 | 78.91 | 6.04 |
| w/o BND | 88.24 | 3.05 | 59.49 | 12.80 | 80.45 | 5.55 |
| RGB | 88.24 | 3.13 | 44.41 | 23.05 | 78.31 | 6.48 |
| HSV | 88.66 | 3.04 | 54.78 | 15.35 | 80.60 | 5.60 |
| RGB + Gaussian noise | 88.39 | 3.15 | 65.05 | 11.25 | 80.17 | 5.75 |
| HSV + Gaussian noise | 86.70 | 3.52 | 63.34 | 11.82 | 79.49 | 5.87 |
| w/o Blur & Distort | 87.29 | 3.56 | **65.78** | **10.88** | 80.43 | 5.57 |
| All | **89.46** | **2.86** | 64.67 | 11.21 | **81.86** | **5.12** |

### 4.2.3 Efficiency

We use AugMix/Shen + the Honda/Waymo datasets + ResNet-50 as the baselines for testing the efficiency. Using Intel Core i7-11700K CPU with 32G RAM and Nvidia RTX 3080 GPU, on the Honda dataset, AutoJoin takes 109 seconds per epoch on average, while AugMix takes 118 seconds and Shen takes 759 seconds. Our approach **saves 8% and 86% per epoch time** compared to AugMix and Shen, respectively. On the Waymo dataset, AugMix takes 128 seconds and Shen takes 818 seconds, while AutoJoin takes only 118 seconds – **8% and 86% time reduction** compared to AugMix and Shen, respectively. Note that the time listed for Shen excludes perturbation selection process during training, which adds to overall training time. More time comparisons such as AugMix/Shen on the Sully/A2D2 datasets with the Nvidia backbone can be found in Appendix F.

Moreover, our approach requires **90% less training data** needed by Shen. This is because we perturb the original clean dataset during the training process, unlike Shen's method of creating nine perturbed datasets (one for each base perturbation) beforehand and combining them with the clean dataset.

## 5 Conclusion, Limitations, and Future Work

We propose AutoJoin, a gradient-free adversarial training technique that is simple yet effective and efficient for robust maneuvering. We show that AutoJoin outperforms state-of-the-art adversarial techniques on various real-word driving datasets through extensive experimentation. AutoJoin is the most efficient technique tested on by being faster per epoch compared to AugMix and saving 83% per epoch time and 90% training data over Shen

AutoJoin is constrained to the regression task of autonomous driving rather than being a general-purpose data augmentation technique. Our focus is on autonomous driving given the significance in making autonomous driving systems robust to perturbations considering the real-world impacts (such as accidents or fatalities) that could occur if such driving systems fail. Another limitation is that while we use real-world images for training/testing, we have not deployed AutoJoin in a real-world test driving scenario. This limits our understanding of how AutoJoin would work within day-to-day driving scenarios. Such testing would be absolutely necessary in order to ensure the efficacy of AutoJoin in autonomous driving systems and the safety of all parties during the testing scenario.

In the future, we are interested in expanding AutoJoin to explore wider perturbation space and more intensity levels to remove any use of FID as well as using other perturbation sets. Furthermore, we would like to explore means to improve clean and combined performance on the Honda/Waymo datasets with ResNet-50. Although our work lacks theoretical support, this remains an open problem since the state-of-the-art techniques we compared with also lack theoretical evidence. Seeking theoretical support for our findings would be an interesting research direction.

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
