

Figure 3: Sample images used during training within the SullyChen (Chen, 2017) dataset. The clean image and its perturbed variants using all base perturbations are shown. The intensity level of the images is 0.5, half of the max intensity.

## A  DATASETS AND EXPERIMENT SETUP

**Base Perturbations.** The description of the base perturbations is given in Sec. 3.1. As an example, in Fig. 3 we show the clean image and the perturbed images from all base perturbations. The perturbation intensity is 0.5, half of the maximum intensity.

**Driving datasets and perturbed datasets.** We use four driving datasets in our experiments: Honda (Ramanishka et al., 2018), Waymo (Sun et al., 2020), A2D2 (Geyer et al., 2020), and SullyChen (Chen, 2017). Theses datasets have been widely adopted for developing machine learning models for driving-related tasks (Xu et al., 2019; Shi et al., 2020; Yi et al., 2021; Shen et al., 2021). Based on these four datasets, we generate test datasets that contain more than 5M images in six categories. Four of them are gradient-free, named Clean, Single, Combined, Unseen, and are produced according to Shen to ensure fair comparisons. We also present details for two gradient-based datasets, FGSM and PGD, which are used to test our approach's adversarial transferability in Appendix G.

- Clean: the original driving datasets Honda, Waymo, A2D2, and SullyChen.
- Single: images with a single perturbation applied at five intensity levels from Shen over the 15 perturbations introduced in Sec. 3.1. This results in 75 test cases in total.
- Combined: images with multiple perturbations at the intensity levels drawn from Shen. There are six combined test cases in total.
- Unseen: images perturbed with simulated effects, including fog, snow, rain, frost, motion blur, zoom blur, and compression, from ImageNet-C (Hendrycks and Dietterich, 2019). Each effect is perturbed at five intensity levels for a total of 35 unseen test cases.
- FGSM: adversarial images generated using FGSM (Goodfellow et al., 2014) with either the Nvidia model or ResNet-50 trained only on clean data. FGSM generates adversarial examples in a single step by maximizing the gradient of the loss function with respect to the images. We generate test cases within the bound $L_\infty$ norm at five step sizes $\epsilon = 0.01, 0.025, 0.05, 0.075$ and $0.1$.
- PGD: adversarial images generated using PGD (Madry et al., 2017) with either the Nvidia model or ResNet-50 trained only on clean data. PGD extends FGSM by taking iterative steps to produce an adversarial example at the cost of more

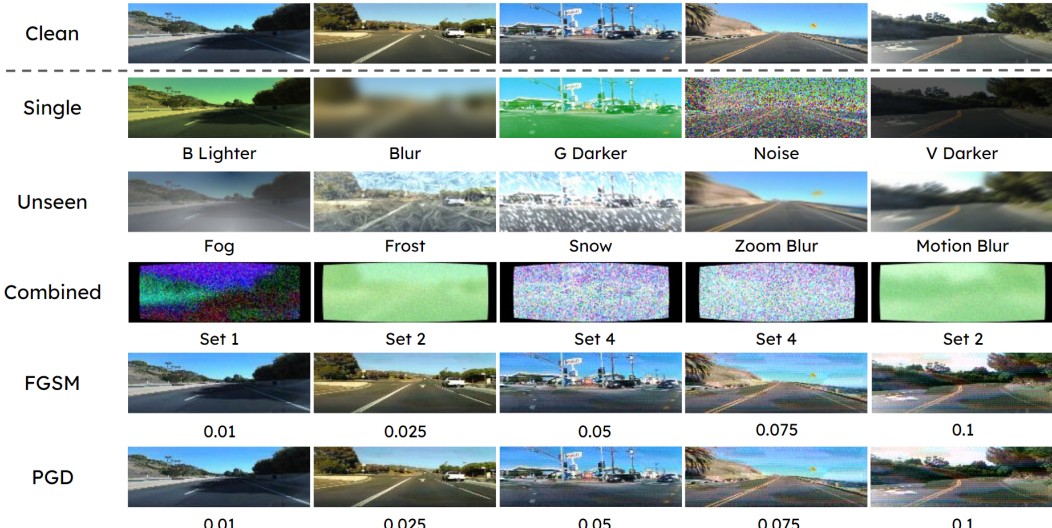

Figure 4: Sample images with perturbations for the six test categories. A column represents a single image that is either clean or perturbed by one of the five perturbation categories. Single images are perturbed by only one of the perturbations outlined in Sec. 3.1 Unseen images contain corruptions from ImageNet-C (Hendrycks and Dietterich, 2019). Combined images have multiple perturbations overlaid, for example, the second column image has G, noise, and blur as the most prominent perturbations. FGSM and PGD adversarial examples are also shown at increasing intensities. The visual differences are not salient due to the preservation of gradient-based adversarial attack potency.

computation. Again, we generate test cases at five intensity levels with the same max bounds as of FGSM.

Sample images for each test category are given in Fig. 4. For Single and Unseen, perturbed images were selected with intensities from 0.5 to 1.0 to highlight the perturbation.

**Network architectures.** We test on two backbones, the Nvidia model (Bojarski et al., 2016) and ResNet-50 (He et al., 2016). We empirically split the Nvidia model where the encoder is the first seven layers and the regression head is the last two layers; for ResNet-50, the encoder is the first 49 layers and the regression head is the last fully-connected layer. The decoder is a five-layer network with ReLU activations between each layer and a Sigmoid activation for the final layer.

**Performance metrics, computing platforms, and training parameters.** We evaluate our approach using mean accuracy (MA) and mean absolute error (MAE). MA is defined as $\sum_\tau acc_{\tau \in \mathcal{T}}/|\mathcal{T}|$ where $acc_\tau = count(|a_p - a_t| < \tau)/n$, where $n$ denotes the number of test cases, $\mathcal{T} = \{1.5, 3.0, 7.5, 15.0, 30.0, 75.0\}$, and $a_p$ and $a_t$ are the predicted angle and true angle, respectively. Note that we do not use the AMAI/MMAI metrics, which are derived from MA scores, from Shen et al. (2021) since AMAI/MMAI only show performance improvement while the actual MA scores are more comprehensive. All experiments are conducted using Intel Core i7-11700k CPU with 32G RAM and Nvidia RTX 3080 GPU. We use the Adam optimizer (Kingma and Ba, 2014), batch size 124, and learning rate $10^{-4}$ for training. All models are trained for 500 epochs.

# B   MAXIMUM AND MINIMUM INTENSITY

We examine the effects of using different ranges of intensities for the perturbations. The original range of intensities for AutoJoin is $[c_{min} = 0, c_{max} = 1)$. We perform two sets of experiments: 1) we change $c_{max}$ to be one of $\{0.9, 1.1, 1.2, 1.3, 1.4, 1.5\}$ while leaving $c_{min} = 0$; and 2) we change $c_{min}$ to be one of $\{0.1, 0.2, 0.3, 0.4, 0.5\}$ while leaving $c_{max} = 1$. For the first experiment set, we change $c_{max}$ to be primarily greater than one to see if learning on more intense perturbations allows for better performance. We also change $c_{max}$

Table 7: Comparison results on the SullyChen dataset with the Nvidia model using a different range of intensities. The first results set show using a different maximum intensity value, leaving the minimum value at zero. The second results set show using a different minimum value, leaving the maximum value as one. For both sets, the original range of AutoJoin achieves the best overall performance.

| | Clean | | Single | | Combined | | Unseen | |
|---|---|---|---|---|---|---|---|---|
| | MA (%) | MAE | MA (%) | MAE | MA (%) | MAE | MA (%) | MAE |
| Max 0.9 | 88.66 | 3.09 | 84.64 | 4.46 | 67.77 | 10.11 | 81.01 | 5.35 |
| Max 1.1 | 88.90 | 3.03 | 85.50 | 4.14 | 67.20 | 10.44 | 81.82 | 5.24 |
| Max 1.2 | 87.77 | 3.29 | 84.47 | 4.32 | **67.88** | **9.88** | 80.83 | 5.43 |
| Max 1.3 | 87.92 | 3.30 | 84.70 | 4.33 | 67.87 | 9.94 | 81.22 | 5.33 |
| Max 1.4 | 88.07 | 3.24 | 84.95 | 4.29 | 67.44 | 10.15 | 81.16 | 5.37 |
| Max 1.5 | 87.74 | 3.24 | 84.56 | 4.29 | 65.57 | 10.85 | 80.98 | 5.39 |
| Min 0.1 | 88.60 | 3.10 | 85.33 | 4.14 | 67.78 | 10.01 | 80.97 | 5.50 |
| Min 0.2 | 87.14 | 3.46 | 83.95 | 4.54 | 66.57 | 10.49 | 80.05 | 5.74 |
| Min 0.3 | 87.14 | 3.31 | 84.27 | 4.32 | 66.35 | 10.60 | 80.49 | 5.44 |
| Min 0.4 | 87.41 | 3.23 | 84.18 | 4.34 | 66.18 | 10.65 | 80.50 | 5.50 |
| Min 0.5 | 87.56 | 3.33 | 84.20 | 4.41 | 65.58 | 10.87 | 80.14 | 5.62 |
| AutoJoin | **89.46** | **2.86** | **86.90** | **3.53** | 64.67 | 11.21 | **81.86** | **5.12** |

to 0.9 to see if the model does not have to learn on the full range defined by Shen et al. (2021) and still achieves good performance. For the second experiment set, we increase the minimum to see if it is sufficient to learn on images with either no perturbation or a low intensity perturbation to achieve good performance.

Table 7 shows the full set of results for SullyChen using the Nvidia architecture. When changing $c_{max}$, the value of 1.1 achieves the most similar performance compared to the original range of AutoJoin; however, it still performs worse than the original range overall. When looking at changing $c_{min}$, the value of 0.1 results in the closest performance to the original range; however, it also fails to outperform the original range. Looking at both sets of results, changing either $c_{min}$ or $c_{max}$ tends to result in the same magnitude of worse performance for the Clean and Single test categories. However, they differ in that changing $c_{min}$ results in worse performance overall in Unseen for both MA and MAE. These results show a potential vulnerability of the original range as they all outperform the original range in Combined with $c_{max}$ being equal to 1.2 showing the best performance in that category. The results for changing the maximum value show that it is not necessarily the case that learning on more intense perturbations will lead to overall better performance. This could be because the perturbations become intense enough that information necessary for steering angle prediction is lost. The results for changing the minimum value show that it is important for the model to learn on images with no perturbation or a low intensity perturbation given that a minimum of 0.1 achieves the best performance within the set. Overall, however, the original range of AutoJoin achieves the best prediction performance.

The results for A2D2, shown in Table 8, are more inconsistent than SullyChen given that the original range is outperformed in four columns instead of just two. The original range is outperformed by changing $c_{max}$ to 1.3 for the Clean MA and Unseen MA columns and changing $c_{max}$ to 1.4 for Combined. Table 8 does show, similar to Table 7, that learning on more intense perturbations will necessarily lead to better performance when the test intensities are left unchanged. These results also show it is important to learn on images without a perturbation/low intensity perturbation given that the original range outperforms all of the experiments when changing the minimum value. When examining both Table 7 and Table 8, the only times the new ranges outperform the original range of AutoJoin is when $c_{max}$ is increased. However, for both SullyChen and A2D2, the original range achieves the best overall performance.

Table 8: Comparison results on the A2D2 dataset with the Nvidia model using a different range of intensities. The first results set show using a different maximum intensity value, leaving the minimum value at zero. The second results set show using a different minimum value, leaving the maximum value as one. For both sets, the original range of AutoJoin achieves the best overall performance.

| | Clean | | Single | | Combined | | Unseen | |
|---|---|---|---|---|---|---|---|---|
| | MA (%) | MAE | MA (%) | MAE | MA (%) | MAE | MA (%) | MAE |
| Max 0.9 | 83.82 | 7.16 | 82.13 | 7.65 | 74.06 | 9.79 | 79.04 | 8.79 |
| Max 1.1 | 83.95 | 7.17 | 82.78 | 7.54 | 78.51 | 8.87 | 79.59 | 8.63 |
| Max 1.2 | 84.50 | 7.12 | 83.34 | 7.45 | 79.28 | 8.50 | 80.28 | 8.62 |
| Max 1.3 | **84.90** | 7.01 | 83.60 | 7.38 | 79.37 | 8.65 | **80.63** | 8.38 |
| Max 1.4 | 84.53 | 7.06 | 83.48 | 7.36 | **79.63** | **8.41** | 80.59 | 8.26 |
| Max 1.5 | 84.65 | 6.86 | 83.39 | 7.26 | 79.50 | 8.57 | 80.37 | 8.30 |
| Min 0.1 | 83.74 | 7.04 | 82.22 | 7.61 | 73.89 | 10.84 | 79.32 | 8.71 |
| Min 0.2 | 84.29 | 7.17 | 83.19 | 7.50 | 77.70 | 9.28 | 79.91 | 8.69 |
| Min 0.3 | 84.16 | 7.35 | 83.12 | 7.61 | 77.16 | 9.16 | 79.77 | 8.82 |
| Min 0.4 | 83.80 | 7.33 | 82.82 | 7.60 | 77.00 | 9.20 | 79.17 | 8.98 |
| Min 0.5 | 84.06 | 7.41 | 82.93 | 7.72 | 75.89 | 10.17 | 79.33 | 9.09 |
| AutoJoin | 84.70 | **6.79** | **83.70** | **7.07** | 79.12 | 8.58 | 80.31 | **8.23** |

Table 9: Results on the SullyChen dataset with the Nvidia backbone and including the feedback loop. The weight coefficients are presented in the order of terms of Eq. 2. Given the overall decreased performance, we exclude the feedback loop from AutoJoin.

| | Clean | | Single | | Combined | | Unseen | |
|---|---|---|---|---|---|---|---|---|
| | MA (%) | MAE | MA (%) | MAE | MA (%) | MAE | MA (%) | MAE |
| (10,1,1) | 86.93 | 3.56 | 83.60 | 4.66 | 64.10 | 11.10 | 78.88 | 6.25 |
| (1,10,1) | 89.11 | 3.07 | 85.60 | 4.15 | **68.23** | **9.70** | 81.58 | 5.27 |
| (1,1,10) | 81.34 | 5.08 | 78.77 | 6.04 | 64.36 | 11.21 | 76.31 | 6.71 |
| AutoJoin (1,10) | **89.46** | **2.86** | **86.90** | **3.53** | 64.67 | 11.21 | **81.86** | **5.12** |

## C  FEEDBACK LOOP

This section contains results and discussion for the SullyChen, A2D2, Honda, and Waymo datasets. Adding the denoised images results in adding a third term to Eq. 1.

$$\mathcal{L} = \lambda_1 \ell_2 \left( \mathbf{x'_i}, \mathbf{x_i} \right) + \lambda_2 \ell_1 \left( \mathbf{a_{p_i}}, \mathbf{a_{t_i}} \right) + \lambda_3 \ell_1 (\mathbf{a_{p'_i}}, \mathbf{a_{t_i}}), \tag{2}$$

where $\lambda_3$ is the weight of the new term and $a_{p'_i}$ is the predicted steering angle on the reconstruction $x'_i$.

Looking at Table 9, emphasizing the reconstruction regression loss causes significant performance loss compared to AutoJoin with 8.12% MA/2.22 MAE and 8.13% MA/2.51 MAE decreases on Clean and Single, respectively. This suggests the data contained within the reconstructions is detrimental to the overall performance/robust capabilities of the regression model. Emphasizing reconstruction loss results in worse performance than AutoJoin, which is expected as AutoJoin emphasizes regression loss for the SullyChen dataset. Emphasizing the regression loss results in improvements in Combined (3.56% MA and 1.51 MAE) at the cost of detriment to performance in all other categories. Overall, there is a decrease in performance when adding the feedback loop.

In Table 10, a similar trend to Table 9 is seen where the weight tuple (1,10,1) is the best performing of the three weight tuples, while the tuple (1,1,10) offers the worst performance. This reiterates that emphasizing the reconstruction regressions had detrimental effects, which

Table 10: Comparison results on the A2D2 dataset with the Nvidia model using different subsets of the original set of perturbations. The weight coefficients are presented in the order: reconstruction loss, regression loss, reconstruction regression loss. AutoJoin's original set of weights outperforms all three weight coefficient tuples.

|  | Clean | | Single | | Combined | | Unseen | |
|---|---|---|---|---|---|---|---|---|
|  | MA (%) | MAE | MA (%) | MAE | MA (%) | MAE | MA (%) | MAE |
| (10,1,1) | 83.98 | 7.07 | 82.81 | 7.44 | 78.63 | 8.75 | 79.26 | 8.95 |
| (1,10,1) | 84.18 | 7.05 | 83.09 | 7.38 | 77.80 | 8.81 | 79.72 | 8.57 |
| (1,1,10) | 83.01 | 7.42 | 81.89 | 7.78 | 77.21 | 8.97 | 79.67 | 8.39 |
| AutoJoin (1,10) | **84.70** | **6.79** | **83.70** | **7.07** | **79.12** | **8.58** | **80.31** | **8.23** |

Table 11: Comparison results on the Honda dataset with the ResNet-50 model using different subsets of the original set of perturbations. The weight coefficients are presented in the order: reconstruction loss, regression loss, reconstruction regression loss. Adding the feedback loop for the Honda dataset, results in significant performance loss for all three weight tuples. Because of this, the original weights of AutoJoin achieve the best performance.

|  | Clean | | Single | | Combined | | Unseen | |
|---|---|---|---|---|---|---|---|---|
|  | MA (%) | MAE | MA (%) | MAE | MA (%) | MAE | MA (%) | MAE |
| (10,1,1) | 79.58 | 6.53 | 77.19 | 8.13 | 65.19 | 18.21 | 77.15 | 8.84 |
| (1,10,1) | 85.70 | 3.53 | 83.23 | 4.67 | 61.41 | 20.68 | 81.13 | 5.72 |
| (1,1,10) | 51.30 | 14.98 | 49.19 | 16.79 | 42.71 | 22.15 | 49.15 | 17.15 |
| AutoJoin (1,10) | **96.46** | **1.12** | **94.58** | **1.98** | **70.70** | **14.56** | **91.92** | **2.89** |

is potentially due to information loss within the reconstructions. However, A2D2 is less affected by adding an additional loss term compared to SullyChen. This is seen as the differences in ranges of performance between the weight coefficients are much greater for SullyChen than for A2D2. For example, the range of performance on Clean for SullyChen is 7.77% MA/2.01 MAE while for A2D2, it is just 1.17% MA/0.37 MAE. Overall, the original weights of AutoJoin provide for the best performance.

Table 11) shows the results for Honda with ResNet-50. The results show that adding the feedback loop for Honda results in significant performance loss, even when emphasizing regression loss. For example, the greatest differences between the three experiment weight tuples and AutoJoin's original weight tuple are 8.12% MA and 2.22 MAE for SullyChen and 1.91% MA and 0.71 MAE for A2D2. However, the least differences between the three weight tuples and AutoJoin's original weight tuple for Honda is 9.29% MA and 2.41 MAE. Emphasizing the reconstruction regression loss results in significant performance losses of 45.16% MA and 13.86 MAE in Clean, which is a 47% decrease for MA and a 93% decrease for MAE; Single, Combined, and Unseen also have significant performance losses. Emphasizing regression loss also results in significant performance loss such as 11.35% MA and 2.69 MAE decreases in Single; these equates to a 14% MA decrease and 57% MAE decrease. These significant performance losses are part of our reasoning to exclude the feedback loop as a main component of AutoJoin. AutoJoin performs better on Honda without the feedback loop.

Table 12 shows the results for adding the feedback loop to Waymo on ResNet-50. Waymo uses a weight tuple of (10,1) in AutoJoin for better performance, while the other datasets use (1,10). This suggests that learning on the underlying distribution of the data and the reconstructions provide significant benefit over emphasizing regression loss. However, when adding the feedback loop, emphasizing the regression loss results in better performance than emphasizing the reconstruction loss; however, both are outperformed by AutoJoin. The performance trend for Waymo is significantly different from the other datasets as emphasizing the reconstruction regression loss results in SOTA performance. AutoJoin-Fuse's results are shown for further comparison since it is the SOTA within the main text. This is an

Table 12: Comparison results on the Waymo dataset with the ResNet-50 model using different subsets of the original set of perturbations. The weight coefficients are presented in the order: reconstruction loss, regression loss, reconstruction regression loss. Emphasizing the reconstruction regression loss term results in SOTA performance.

| | Clean | | Single | | Combined | | Unseen | |
|---|---|---|---|---|---|---|---|---|
| | MA (%) | MAE | MA (%) | MAE | MA (%) | MAE | MA (%) | MAE |
| (10,1,1) | 63.14 | 19.15 | 63.38 | 19.11 | 57.98 | 23.15 | 61.61 | 20.41 |
| (1,10,1) | 63.35 | 18.89 | 63.22 | 19.37 | 56.32 | 35.88 | 62.32 | 21.31 |
| (1,1,10) | **67.70** | 18.00 | **66.68** | **18.28** | **67.70** | **18.00** | **67.70** | **18.00** |
| AutoJoin (10,1) | 64.91 | 18.02 | 63.84 | 19.30 | 58.74 | 26.42 | 64.17 | 19.10 |
| Fuse (10,1) | 65.07 | **17.60** | 64.34 | 18.49 | 63.48 | 20.82 | 65.01 | 18.17 |

Table 13: Comparison results on the SullyChen dataset with the Nvidia model looking at the cases where it is not guaranteed the **F**ull **S**et of perturbations is seen by the model, not using **R**andom **I**ntensities, or both. The distinct intensities come from Shen. Using the original AutoJoin setup results in the best overall performance across all subsets of perturbations.

| | Clean | | Single | | Combined | | Unseen | |
|---|---|---|---|---|---|---|---|---|
| | MA (%) | MAE | MA (%) | MAE | MA (%) | MAE | MA (%) | MAE |
| w/o FS | 87.74 | 3.32 | 84.54 | 4.34 | 66.27 | 10.32 | 80.62 | 5.50 |
| w/o RI | 88.07 | 3.42 | 84.76 | 4.42 | **67.72** | **10.12** | 80.92 | 5.65 |
| w/o FS+RI | 86.43 | 3.54 | 83.19 | 4.62 | 61.97 | 13.01 | 78.51 | 6.23 |
| AutoJoin | **89.46** | **2.86** | **86.90** | **3.53** | 64.67 | 11.21 | **81.86** | **5.12** |

intriguing development because of the negative performance impacts that the feedback loop has on the other datasets. This result is further evidence towards the idea the learning underlying distributions of Waymo leads to better performance. Overall, emphasizing the reconstruction regression results in SOTA performance.

## D FULL SET OF PERTURBATIONS NOT GUARANTEED AND NO RANDOM INTENSITIES

From Table 5, AutoJoin without the denoising autoencoder (DAE) already outperforms the work by Shen et al. (2021). Outside of adding the DAE, the main changes from their work to our work is that we ensure that all 15 perturbations are seen during learning and that the intensities are sampled from a range instead of using distinct intensities. Thus, we want to examine if these changes have an effect on performance and can account for the reason that AutoJoin without the DAE outperforms the Shen model. We break down this set of experiments into three sets of cases: 1) not guaranteeing the full set of 15 perturbations are seen by the model, 2) not using random intensities, or 3) both. The original methodology of AutoJoin is left the same except for the changes of each case. The third case brings the methodology of AutoJoin closest to that of the work by Shen et al. (2021) although they are not the same entirely.

The first case is accomplished by not discretizing the single channel perturbations as described in Sec. 3.1. Whether to lighten or darken the R, G, B, H, S, and V channels of the images is decided stochastically. This means there is potential the model does not see all 15 perturbations, although highly unlikely; however, it is highly likely that the model does not see them with the same frequency as with the original methodology of AutoJoin. The second case is done by using the five distinct intensities from the work by Shen et al. (2021), which are {0.02, 0.2, 0.5, 0.65, 1.0}. The intensity for a perturbation is still sampled from within this set of values, but it is inherently not as wide of a distribution space compared to the methodology described in Sec. 3.1. The third case combines the changes in procedure outlined above.

Table 14: Comparison results on the A2D2 dataset with the Nvidia model looking at the cases where it is not guaranteed the **F**ull **S**et of perturbations is seen by the model, not using **R**andom **I**ntensities, or both. Using the original AutoJoin setup results in the best overall performance across all subsets of perturbations.

|            | Clean | | Single | | Combined | | Unseen | |
|------------|--------|------|--------|------|----------|------|--------|------|
|            | MA (%) | MAE | MA (%) | MAE | MA (%) | MAE | MA (%) | MAE |
| w/o FS     | 83.93 | 7.24 | 82.77 | 7.52 | 78.60 | 8.90 | 78.45 | 9.78 |
| w/o RI     | 83.90 | 6.95 | 82.68 | 7.35 | 78.20 | 8.72 | 78.38 | 9.20 |
| w/o FS+RI  | 83.90 | 7.10 | 82.85 | 7.42 | 78.45 | 8.63 | 79.01 | 9.05 |
| AutoJoin   | **84.70** | **6.79** | **83.70** | **7.07** | **79.12** | **8.58** | **80.31** | **8.23** |

Table 15: Comparison results on the Honda dataset with the ResNet-50 model looking at the cases where it is not guaranteed the **F**ull **S**et of perturbations is seen by the model, not using **R**andom **I**ntensities, or both. The model without FS results in the best overall performance of the model, which is different from the SullyChen, A2D2, and Waymo datasets.

|            | Clean | | Single | | Combined | | Unseen | |
|------------|--------|------|--------|------|----------|------|--------|------|
|            | MA (%) | MAE | MA (%) | MAE | MA (%) | MAE | MA (%) | MAE |
| w/o FS     | **96.78** | **1.05** | **95.17** | **1.82** | **75.16** | **12.34** | 91.69 | 3.27 |
| w/o RI     | 96.53 | 1.08 | 94.92 | 1.86 | 68.63 | 17.14 | 91.53 | 3.18 |
| w/o FS+RI  | 96.72 | 1.06 | 95.20 | 1.80 | 65.49 | 24.44 | 90.97 | 3.91 |
| AutoJoin   | 96.46 | 1.12 | 94.58 | 1.98 | 70.70 | 14.56 | **91.92** | **2.89** |

Looking at the effects the three cases have on SullyChen and A2D2 using the Nvidia model, the results show that ensuring all 15 perturbations are seen during learning and sampling the intensities does improve overall performance when predicting steering angles and these changes are significant to the training of the model. This gives more credence to why AutoJoin without the DAE is able to outperform Shen.

The effects of the three cases differs between the two datasets. Table 13 shows that using both is able to significantly impact performance on all categories by decreasing performance by an average of 3.20% MA and 1.17 MAE across all test categories for SullyChen. Using distinct intensities allows for significantly better performance on Combined (the model without FS also achieves better performance in this category), but fails to outperform in Clean, Single, and Unseen categories. For A2D2, the overall effect is much less severe as the differences between the three effects and AutoJoin's methodology are in closer proximity with roughly a difference of 1.0% MA and 0.5 MAE. However, the original setup still results in the overall best steering angle prediction performance.

Table 15 shows the results for Honda with ResNet-50. Unlike SullyChen and A2D2, all three cases actually outperform AutoJoin for both Clean and Single. AutoJoin is even outperformed in Combined when not ensuring the full set. Not ensuring the full set has potential for more variability of when perturbations are learned by the model, which can increase the perturbation distribution space allowing for better generalization. However, when not ensuring the full set and using distinct intensities, there is a loss of generalization as AutoJoin outperforms this case in Combined and Unseen. The Shen model outperforms AutoJoin in Clean and Combined MAE. Shen still outperforms the case of not ensuring the full set on Clean, but the Shen model is outperformed on Combined MAE. AutoJoin achieves the best performance in Unseen amongst all the three cases; however, overall the model without the full set provides for the best performance.

The idea that ensuring all 15 perturbations are seen during learning and sampling the perturbation intensities does improve the overall performance of the model returns with Waymo on ResNet-50. Table 16 shows these results. AutoJoin outperforms all three cases in Clean, Single MA, and Unseen. It is outperformed by the all three cases in Single MAE

Table 16: Comparison results on the Waymo dataset with the ResNet-50 model looking at the cases where it is not guaranteed the **F**ull **S**et of perturbations is seen by the model, not using **R**andom **I**ntensities, or both. Using all of them results in the best overall performance across all subsets of perturbations.

|            | Clean | | Single | | Combined | | Unseen | |
|------------|--------|-------|--------|-------|----------|--------|--------|-------|
|            | MA (%) | MAE | MA (%) | MAE | MA (%) | MAE | MA (%) | MAE |
| W/o FS     | 63.95 | 18.40 | 63.40 | 18.85 | **61.04** | **21.46** | 63.40 | 19.15 |
| W/o RI     | 64.12 | 18.57 | 63.76 | 19.14 | 54.80 | 32.87 | 62.27 | 21.52 |
| W/o FS+RI  | 63.96 | 18.51 | 63.70 | **18.76** | 56.25 | 26.81 | 62.79 | 19.81 |
| AutoJoin   | **64.91** | **18.02** | **63.84** | 19.30 | 58.74 | 26.42 | **64.17** | **19.10** |

Table 17: Comparison results on the A2D2 dataset with the Nvidia model using different subsets of the original set of perturbations. "No BND" means that blur, noise, and distort are not used within the perturbation set. The single perturbation column is removed for a fair comparison. Using all of them results in the best overall performance across all subsets of perturbations.

|                  | Clean | | Combined | | Unseen | |
|------------------|--------|------|----------|-------|--------|------|
|                  | MA (%) | MAE | MA (%) | MAE | MA (%) | MAE |
| No RGB           | 84.46 | 6.61 | 77.08 | 9.51 | 79.67 | 8.65 |
| No HSV           | 84.50 | 6.70 | 76.51 | 9.41 | 78.01 | 9.36 |
| No BND           | 83.72 | 7.21 | 68.88 | 12.09 | 78.49 | 9.13 |
| RGB              | 83.82 | 7.20 | 67.49 | 12.40 | 76.93 | 9.87 |
| HSV              | 83.19 | 7.26 | 67.91 | 12.65 | 78.00 | 9.31 |
| Only RGB+Noise   | 83.92 | 6.91 | 73.56 | 10.03 | 78.44 | 8.96 |
| Only HSV+Noise   | 84.39 | 6.87 | 70.96 | 11.47 | 79.53 | 8.61 |
| No Blur,Distort  | 82.53 | 7.49 | 74.99 | 9.53 | 77.40 | 9.57 |
| All              | **84.70** | **6.79** | **79.12** | **8.58** | **80.31** | **8.23** |

and is outperformed by the model without FS in Combined; however, it outperforms the other two cases in Combined. Looking at the results for Honda and Waymo, it appears that not ensuring all 15 perturbations are seen during training provides for the best performance for Combined; however still fails to outperform AutoJoin in Unseen. These two categories are different from Clean and Single as the model never learns on them during training. The model without FS is able to generalize better for Combined than Unseen given the results.

## E    PERTURBATION STUDY

This section contains more results and discussion for the other datasets of A2D2, Honda, and Waymo for the experiments where different subsets of perturbations are used.

The trends for A2D2 using the Nvidia model are different compared to SullyChen. The results are given in Table 17. While A2D2 is similar to SullyChen in that the best performance comes from using all of the perturbations, the model with no BND perturbations performs the worst on Combined implying that some combination of these perturbations is important for model generalizability for Combined. This idea is aided by the other scenarios where performance on Combined is improved when Gaussian noise is added back to the set of perturbations seen by the model. The closest in overall performance to using all perturbations is not using RGB perturbations within the training set. For Unseen, there are no clear patterns within the performances amongst the various subsets with the worst performing subset being not using blur and distort perturbations at 77.40% MA and 9.57 MAE. The other trend that is similar to SullyChen, however, is that Combined contains the most volatility in the performance; the range from the worst performing subset to the best performing subset is 68.88% MA and

Table 18: Comparison results on the Honda dataset with the ResNet-50 model using different subsets of the original set of perturbations. "No BND" means that blur, noise, and distort are not used within the perturbation set. The single perturbation column is removed for a fair comparison. Using all perturbations is overall the best performing model despite being outperformed in the Clean and Combined categories.

| | Clean | | Combined | | Unseen | |
|---|---|---|---|---|---|---|
| | MA (%) | MAE | MA (%) | MAE | MA (%) | MAE |
| No RGB | **96.78** | **1.02** | 66.37 | 21.67 | 91.72 | 3.26 |
| No HSV | 96.48 | 1.07 | **74.96** | **13.19** | 88.25 | 5.67 |
| No BND | 96.08 | 1.27 | 63.88 | 18.97 | 90.58 | 3.55 |
| RGB | 95.75 | 1.38 | 44.90 | 32.36 | 83.53 | 7.32 |
| HSV | 95.94 | 1.31 | 53.99 | 29.83 | 83.53 | 7.59 |
| Only RGB+Noise | 96.39 | 1.13 | 69.21 | 14.48 | 87.13 | 5.70 |
| Only HSV+Noise | 96.47 | 1.12 | 69.08 | 14.72 | 91.77 | 2.93 |
| No Blur,Distort | 96.33 | 1.17 | 67.74 | 14.73 | 91.16 | 3.15 |
| All | 96.46 | 1.12 | 70.70 | 14.56 | **91.92** | **2.89** |

Table 19: Comparison results on the Waymo dataset with the ResNet-50 model using different subsets of the original set of perturbations. "No BND" means that blur, noise, and distort are not used within the perturbation set. The single perturbation column is removed for a fair comparison. Using all of them results in the best overall performance across all subsets of perturbations.

| | Clean | | Combined | | Unseen | |
|---|---|---|---|---|---|---|
| | MA (%) | MAE | MA (%) | MAE | MA (%) | MAE |
| No RGB | 64.63 | 18.20 | 60.38 | 24.79 | 63.63 | 19.72 |
| No HSV | 63.94 | 18.46 | **61.18** | **20.89** | 63.09 | 20.02 |
| No BND | 64.56 | 18.06 | 50.21 | 43.28 | 63.06 | 20.13 |
| RGB | 64.64 | 18.12 | 49.32 | 36.31 | 62.27 | 20.25 |
| HSV | **65.00** | 18.37 | 52.03 | 34.21 | 63.85 | 19.52 |
| Only RGB+Noise | 63.95 | 18.40 | 52.84 | 31.01 | 60.70 | 23.43 |
| Only HSV+Noise | 64.48 | 17.97 | 59.97 | 24.39 | 63.65 | 19.37 |
| No Blur,Distort | 64.04 | 18.06 | 57.29 | 28.48 | 62.89 | 19.91 |
| All | 64.91 | **18.02** | 58.74 | 26.42 | **64.17** | **19.10** |

12.09 MAE to 79.12% MA and 8.58 MAE. Using all perturbations during learning results in the best performance.

Table 18 shows the results for Honda on ResNet-50. Using all perturbations is outperformed several cases in Clean and Combined. Not using RGB perturbations achieves the best performance in Clean and not using HSV perturbations achieves the best performance in Combined (by a significant margin of 4.26% MA/1.37 MAE). Even with the clean performance increases, the Shen model is still the best in Clean. Well-defined patterns are still not clear in the results. Not using RGB perturbations performs worse than using all perturbations in Combined. Not using HSV perturbations significantly improves performance in Combined at a 4.26% MA and 1.37 MAE improvement; however, results in a significant decrease in performance in Unseen with 3.67% MA and 2.78 MAE detriments. The range of performance for Combined is the largest compared to the other categories. This is similar to SullyChen and A2D2, showing further evidence of the volatility within Combined. The closest in performance to using all perturbations is not using HSV perturbations; this case results in a net gain of 0.61% MA and a net loss of 1.36 MAE when compared to using all. Given that MAE, for Honda, are small values and lie within a tighter range than MA, the net loss of 1.36 MAE means that using all perturbations is actually the best performing model overall.

Table 20: Table comparing the efficiency of different techniques in terms of time (in seconds) per each epoch on the ResNet-50 model. AutoJoin is the most efficient out of all the techniques.

|  | Honda | Waymo |
| --- | --- | --- |
| Standard | 90 | 97 |
| AugMix | 118 | 128 |
| Shen | 759 | 818 |
| AutoJoin | **109** | **118** |

Table 21: Table comparing the efficiency of different techniques in terms of time (in seconds) per each epoch on the Nvidia model. AutoJoin is the most efficient out of all the techniques.

|  | SullyChen | A2D2 |
| --- | --- | --- |
| Standard | 2 | 4 |
| AugMix | 10 | 22 |
| Shen | 9 | 16 |
| AutoJoin | **5** | **9** |

Table 19 shows the results for Waymo with ResNet-50. Using all perturbations results in the best overall performance for the model, although not using HSV perturbations outperforms using all in Combined for both MA and MAE. Combined has the widest range in performances amongst the subsets confirming that Combined, in general, is the most volatile in performance across all the datasets used. Not using RGB perturbations, not using HSV perturbations, and only using HSV perturbations and Gaussian noise outperform using all perturbations in Combined; however, this does not translate over to Clean and Unseen. Only using RGB and Gaussian noise perturbations results in the overall worst performance across the three categories, but any further patterns can not be well-defined from this as using RGB perturbations and/or Gaussian noise in other cases results in relatively good performance. Overall, using all perturbations results in the best performing prediction model.

## F  TIMES FOR EXPERIMENTS

We present Tables 20 and 21 with times for various experiments. Table 20 shows time (in seconds) per epoch for Standard, AugMix, Shen, and AutoJoin The experiments are on AutoJoin, Shen, AugMix, and Standard with both the ResNet-50 and Nvidia models. Standard is given to show baseline efficiency when not performing any robustness training. From the table, AutoJoin is the most efficient compared to the other techniques. AugMix is close in efficiency as it is within at most 10 seconds of our technique's time. Shen's efficiency is significantly worse compared to both AutoJoin and AugMix as it is many times slower than both techniques. Note that for both tables, Shen's time does not reflect a selection process that occurs in-between training that results in additional training time.

Table 21 shows the times for Standard, AugMix, Shen, and AutoJoin using the Nvidia model. Standard is given to provide a baseline efficiency when not performing any robustness training. AutoJoin still achieves the best efficiency out of the techniques; however, Shen is more efficient than AugMix, which is not the case in Table 20. This suggests that with smaller datasets, Shen's technique is able to maintain efficiency and as the datasets starts growing, Shen's efficiency significantly decreases.

## G  GRADIENT-BASED ADVERSARIAL TRANSFERABILITY

Although AutoJoin is a gradient-free technique with the focus on gradient-free attacks, we curiously test it on gradient-based adversarial examples. Dataset details and sample images

Table 22: Results on gradient-based adversarial examples using the A2D2 dataset and the Nvidia backbone. Each column represents a dataset generated at a certain intensity of FGSM/PGD (higher values mean higher intensities). All results are in MA (%). AutoJoin achieves the least adversarial transferability among all techniques tested under all intensities of FGSM (Goodfellow et al., 2014) and PGD (Madry et al., 2017).

| | FGSM | | | | | PGD | | | | |
|---|---|---|---|---|---|---|---|---|---|---|
| | 0.01 | 0.025 | 0.05 | 0.075 | 0.1 | 0.01 | 0.025 | 0.05 | 0.075 | 0.1 |
| Standard | 73.91 | 65.42 | 57.70 | 53.27 | 50.12 | 73.87 | 65.60 | 57.93 | 53.43 | 51.07 |
| AdvBN | 76.34 | 76.14 | 75.50 | 74.25 | 72.75 | 76.35 | 76.17 | 75.62 | 74.46 | 72.91 |
| AugMix | 77.66 | 76.69 | 73.61 | 69.74 | 66.38 | 77.65 | 76.75 | 73.74 | 69.75 | 66.40 |
| AugMax | 77.04 | 76.94 | 76.18 | 75.10 | 73.91 | 77.04 | 76.93 | 76.23 | 75.10 | 73.91 |
| MaxUp | 78.71 | 78.47 | 78.10 | 77.42 | 76.71 | 78.71 | 78.47 | 78.09 | 77.39 | 76.72 |
| Shen | 80.10 | 79.83 | 79.02 | 77.94 | 76.98 | 80.09 | 79.79 | 79.02 | 77.93 | 76.94 |
| AutoJoin | **84.11** | **83.83** | **83.13** | **82.02** | **81.14** | **84.14** | **83.84** | **83.15** | **81.97** | **81.09** |

are given in Appendix A The evaluation results using the A2D2 dataset and the Nvidia backbone are shown in Table 22. AutoJoin surprisingly demonstrates superb ability in defending adversarial transferability against gradient-based attacks by outperforming every other approaches by large margins at all intensity levels of FGSM and PGD.