# OpenReview forum: "AutoJoin: Efficient Adversarial Training against Gradient-Free Perturbations for Ro- bust Maneuvering via Denoising Autoencoder and Joint Learning"
_ICLR.cc/2024/Conference — Submitted to ICLR 2024_

### Official Review · Reviewer_qavn · 2023-10-20

**Soundness:** 2 fair
**Presentation:** 2 fair
**Contribution:** 2 fair
**Rating:** 3
**Confidence:** 4

**Summary:**

The paper propsoed a joint training to imporve the roboustness of image-based maneuvering models. The proposed method improve the roboustness of the two modules jointly, the decoder and steering angle prediction model.

**Strengths:**

The paper focuses on an interesting and important issue of deep nueral networks, especially for autnomous driving systems.

**Weaknesses:**

There are several issues with the paper:
1- The issue of adversarial attack is well-known in the field. However, it is not well explained how this issue might be the case for application discussed in the paper. I think it is important to provide real-world scenarios for this purpose.
2- The proposed method has not been evalauted on general benchmarks and as such it is difficulat to understand the effectivenss of the proposed method.
3- There are several new state-of-the-art adversarial defense mechanisms in the field currently, and they are missed to be included in the paper.
4- It is diffult to understand what is the main novelity of the proposed method.

**Questions:**

1- How deoes this problem might take palce in real-world scenarios?
2- How does the proposed method comapred with state-of-the-art adversarial training and defence mechanisms?
3- What is the main novelity of the propsoed method? new perturbation training or proposing a new roboust model for steer angle prediction?

---

> ### Author Response · Authors · 2023-11-20
> **Thank you for the review**
>
> Thank you very much for the feedback. We appreciate it.
>
> We envision this problem taking place in real-world scenarios as a result of faulty camera sensors attached to vehicles using such self-driving systems. We cannot expect those systems to work perfectly, so it's important they be robust to whatever may take place during field runs. While some of the training perturbations used (such as the RGB/HSV perturbations) are less likely to occur, we show that these perturbations can be used to improve robustness to real-world events such as frost or snow clouding the camera sensor’s perception ability.
>
> The main focus of this work is on gradient-free perturbations, as opposed to gradient-based perturbations. Appendix G discusses some preliminary results of AutoJoin in relation to adversarial attack techniques such as FGSM and PGD. Due to the gradient-free focus, we do not compare to state-of-the-art adversarial defense mechanisms as we feel those belong more in the category of gradient-based adversarial attacks as opposed to gradient-free perturbations as defined in our paper.
>
> We understand that AutoJoin is a technique that lacks novelty. AutoJoin employs simpler techniques, such as its use of DAE, within a robust training pipeline to improve robustness on steering angle prediction as it relates to autonomous driving. However, despite its simplicity, we believe that we are addressing a problem that is interesting to a broad audience; and on the technical front, we believe if a problem can be solved using a simpler approach, that approach should be preferred. In this work, we opt for a relatively simple approach while outperforming SOTA techniques by a significant margin.

---

> > ### Comment · Reviewer_qavn · 2023-11-22
> >
> > I think the experimental results are not aligned with the objective provided in the feedback. How does the perturbation can model faulty camera sensors?

---

> > > ### Author Response · Authors · 2023-11-23
> > > **Response to Reviewer qavn**
> > >
> > > We envision that the perturbations applied to the images simulate potential artifacts that may occur within a faulty camera sensor. For example, we apply a noise perturbation to the images to simulate individual artifacts that may appear in images or we apply a blur perturbation to simulate images that appear blurry due to faulty camera sensors.

---

### Official Review · Reviewer_7kUe · 2023-10-30

**Soundness:** 3 good
**Presentation:** 2 fair
**Contribution:** 3 good
**Rating:** 5
**Confidence:** 1

**Summary:**

This paper is on the topic of robust maneuvering in view of autonomous driving. The authors propose to augment the training images with gradient-free perturbations. Furthermore, they propose to regularize the model training by adding a denoising task where a decoder should reconstruct the original images without the gradient-free perturbations.

**Strengths:**

Overall, the paper is clear. A wide range of experiments are conducted to support their claim, including experiments on several datasets, an ablation study, and experiments showing that gradient-free perturbation and denoising auto-encoder regularization are helpful in improving performance.

**Weaknesses:**

More analysis regarding why the proposed method works is important. For high-level tasks, adding a task of reconstructing the input image or its variant usually degrades the performance on clean data.

Besides, the limitations of this paper discussed at ICLR 2013 seem to still exist, and there seem no significant changes from the last submission.

**Questions:**

Please see the weakness part

---

> ### Author Response · Authors · 2023-11-20
> **Thank you for the review**
>
> Thank you for your feedback.
>
> In future revisions, we will investigate further methods regarding why AutoJoin works as it does. We are planning to add more theoretical verifications of why our approach works to further solidify its foundation as a robustness technique for predicting steering angles. We understand the limitations still exist, so we will work to improve AutoJoin to overcome those limitations. Thank you for your time and your review.

---

### Official Review · Reviewer_vA9d · 2023-11-01

**Soundness:** 3 good
**Presentation:** 2 fair
**Contribution:** 3 good
**Rating:** 5
**Confidence:** 3

**Summary:**

This paper aims to improve the perception robustness of a self-driving system. Specifically, this paper propose a joint learning method, to regress the learning target and to reconstruct the clean image at the same time. With solid evaluation, the proposed method can significantly improve the robustness of the model.

**Strengths:**

The experiments of this paper is solid. The experiments are conducted on a number of datasets, and compared with a number of methods to improve model robustness.

This research topic is very interesting, and aligns well with the real world needs. Before this paper, I didn't see many papers working on this topic.

**Weaknesses:**

There is no enough context information about the task this paper working on. Predicting steering angle from camera image seems not a very popular task, so it is worth introduce this a little bit more.

It is very interesting to see the sensitivity of the hyper-parameters. IIUC, if we set the weights of the reconstruction loss as 0, I suppose the performance will regress back to Standard (FSRI). Please correct me if my understanding is wrong.

For equation (1), I am wondering what's the reason why we times $\lambda_1$ with $l_2$, and times $\lambda_2$ with $l_1$? It seems very counter intuitive so would like to learn the reason.

Looks like the proposed method is not specifically design for the steering angle prediction task, so I am wondering why choose this task, and if the propose method can benefit other self-driving related tasks.

**Questions:**

See weakness.

---

> ### Author Response · Authors · 2023-11-20
> **Thank you for your feedback**
>
> We thank you for your review; we appreciate your feedback.
>
> In a future revision of the paper, we will add more context to the driving task to really solidify its importance to the reader. We thank you for this suggestion.
>
> For setting reconstruction loss weight to 0: Yes, the performance will regress back to Standard (FSRI) considering that essentially eliminates the added decoder part to the architecture.
>
> Equation 1 is written as such since this matches what is present within the code; however, we could easily change the equation around such that the subscripts of the λ’s match those of the loss functions as it is the same either way.
>
> We choose the task of predicting steering angles to match the work by Shen et al. as this is the task they use. We wanted to directly be able to compare to Shen et al. so we retained the same task as them. As such, we have not experimented on other self-driving related tasks with AutoJoin. However, that is an exciting research direction so we thank you for the suggestion.

---

### Official Review · Reviewer_gYbZ · 2023-11-05

**Soundness:** 2 fair
**Presentation:** 3 good
**Contribution:** 3 good
**Rating:** 5
**Confidence:** 4

**Summary:**

The paper presents a novel approach to enhancing the robustness of 'perception-to-control' systems, which are increasingly prevalent due to the integration of machine learning algorithms and ubiquitous sensors. The proposed method, AutoJoin, is a gradient-free adversarial training technique specifically designed for image-based maneuvering tasks. AutoJoin stands out from other state-of-the-art methods, demonstrating significant performance improvements when tested on a substantial dataset of over 5 million images. The technique showcases up to a 40% increase in performance against perturbations, while also achieving a remarkable 300% improvement in clean performance. This indicates that AutoJoin not only enhances the system's resilience against adversarial attacks but also boosts its overall effectiveness in standard operating conditions. In terms of efficiency, AutoJoin proves to be highly advantageous, saving up to 86% of the time per training epoch and requiring 90% less training data compared to other leading techniques. This efficiency makes AutoJoin a practical choice for real-world applications, where resources and time are often limited. The core innovation of AutoJoin lies in its unique architecture, which incorporates a decoder attachment to the original regression model, resulting in a denoising autoencoder integrated within the system. This design enables the simultaneous and synergistic learning of both maneuvering and denoising sensor input tasks. As a result, the performance of each task is enhanced, leading to a more robust and reliable 'perception-to-control' system. In conclusion, AutoJoin emerges as a groundbreaking technique in the field of adversarial training, offering substantial improvements in both performance and efficiency. Its unique architecture and gradient-free approach make it a promising solution for developing trustworthy 'perception-to-control' systems, capable of operating effectively in diverse and challenging environments.

**Strengths:**

-- AutoJoin is a gradient-free adversarial training technique.

-- AutoJoin  achieves significant performance increases up to the 40% range against perturbations while improving on clean performance up to 300%.

--AutoJoin is also quite efficient in computation cost, saving up to 86% time per training epoch and 90% training data over other state-of-the-art techniques.

-- The paper is well written and easy to follow.

**Weaknesses:**

Main weakness is from how the robustness is evaluated. In verifying the robustness, it would be better to explore more types of perturbations. Such perturbations could be from both common image manipulations and adversarial perturbations. Although several image corruptions are applies, they mainly focus on color perspective. What will happen if other types of image corruptions, such as image rotation, perspective warping, JPEG compression. Such robustness evaluation can be referred from [1]

[1]. Xinhua et al. Only For You: Deep Neural Anti-Forwarding Watermark Preserves Image Privacy. 2023

In addition, such evaluation can also be executed  on adversarial perturbation, such as from FGSM, PGD, Autoattack etc.

Several minor issue:

There are some typos need to pay attention. For instance, the first sentence in introduction, "have"-->"has".

In the conclusion section, the citation is incomplete. Line 5: a full stop is missing at the end of a sentence.

**Questions:**

-- Adversarial training is largely dependent on how the images are perturbed, which determines how the adversarially trained model will generalize its robustness. The FID is used only minimally to determine the maximum intensity value of a perturbation. What will happen or be expected if perturbations are kept consistent with [Shen et al, 2021] and [Bengio et al 2009]?

-- Given the mentioned possible add-on evaluations, the transferability of claimed adversarial robustness can be further tested?

---

> ### Author Response · Authors · 2023-11-20
> **Thank you for your feedback**
>
> Thank you for the review.
>
> The gradient-free perturbations we use during training focus on color perspective; however, we do test on other types of image perturbations such as blur or compression. We had not considered adding those types of image perturbations to the training pipeline to stay inline with Shen et al. so we do not currently have empirical results for what would happen if we used other types of image perturbations during training.
>
> We do investigate AutoJoin in relation to FGSM and PGD adversarial attacks; however, we place that section within Appendix G due to our main focus for the paper being on gradient-free perturbations.
>
> For AutoJoin’s performance with using perturbation intensity values that match Shen et al., we investigate that in Appendix D with the model labeled as “w/o RI” (meaning without Random Intensities) within Tables 14 to 17. Our experiments find that using the same perturbation values as Shen et al., there is a loss in performance overall across the datasets compared to random intensities sampled within our defined range. As such, we deploy AutoJoin with random intensities compared to the same intensities as Shen et al. In regard to Bengio et al., we have not considered matching those values so we thank you for the suggestion.
>
> Once again, thank you very much for the suggestions. We have resolved the issues regarding the typos and incomplete citation.

---

### Meta-Review · Area_Chair_xwYy · 2023-12-06

**Metareview:**

This paper introduces AutoJoin, a gradient-free adversarial training technique aimed at producing robust models for image-based maneuvering effectively and efficiently. While the paper is interesting and well-organized, there are concerns about the experimental part that need improvement. Reviewer gYbZ emphasizes the importance of incorporating a broader range of image perturbations and recommends rejecting the paper based on concerns about the fairness of comparisons and experimental settings. Reviewer vA9d raises concerns about the benchmark, baselines, and tasks, suggesting the application of the proposed method to different tasks to demonstrate its effectiveness further. Reviewer 7kUe points out remaining limitations, and Reviewer qavn suggests testing the method in real-world scenarios, evaluating it on a general benchmark, and discussing state-of-the-art adversarial defense mechanisms. The rebuttals didn’t address all the above concerns. Considering these factors, the Area Chair recommends rejecting the paper in the current stage. The AC encourages the authors to enhance their paper by incorporating the valuable suggestions provided by the reviewers and to resubmit the revised paper to the next venue.

**Justification For Why Not Higher Score:**

After the rebuttals, concerns about the experiments persist. None of reviewers agree to accept the paper.

**Justification For Why Not Lower Score:**

NA

---

### Decision · Program_Chairs · 2024-01-16

Reject